# Ensuring Equitable COVID-19 Vaccine Allocation in New Hampshire: The First Eight Months toward a New Era

**DOI:** 10.3390/vaccines10091421

**Published:** 2022-08-29

**Authors:** Taylor D. Selembo, Elizabeth A. Talbot, Christophe T. Courtine, Elizabeth R. Daly, Torane W. Hull, Kirsten J. Durzy

**Affiliations:** 1New Hampshire Department of Health and Human Services, Concord, NH 03301, USA; 2Dartmouth Geisel School of Medicine, Hanover, NH 03744, USA; 3Centers for Disease Control and Prevention (CDC), Atlanta, GA 30333, USA

**Keywords:** COVID-19, public health, vaccine, equity, pandemic, coronavirus, allocation

## Abstract

The global coronavirus disease 2019 (COVID-19) pandemic has been exacerbated by social vulnerabilities and racial disparities, resulting in disproportionate morbidity and mortality that require continued attention to strategies that ensure equitable vaccine allocation. The State of New Hampshire (NH) developed a transparent framework to guide COVID-19 vaccine allocation plans, of which one key component was the allocation of 10% of vaccine supply to disproportionately impacted and highly vulnerable populations, predominantly identified through a national vulnerability index. The process, operational approaches, ethical challenges, and unanticipated consequences resulted in many valuable lessons learned. Equitable allocation of this limited and critical pandemic countermeasure required public understanding and engagement, which was achieved through a publicly available framework that was flexible, resourced using public funds, and widely communicated. Broad partnerships were also critical to addressing disparities in the delivery of vaccine. The lessons learned and described here will facilitate more nimble and equitable jurisdictional responses in future public health emergencies.

## 1. Introduction

The coronavirus disease 2019 (COVID-19) pandemic has resulted in over 589 million infections and over 6 million deaths since the virus first emerged in China in late 2019 [1]. In the U.S., racial and ethnic minority populations have been more vulnerable to infection and severe outcomes with COVID-19. Specifically, the national rate of infection with COVID-19 for American Indians/Alaskan Natives non-Hispanics (AI/AN) and Hispanic or Latino persons was approximately 1.5 times the rate for White Americans [2]. The rates of hospitalization and death nationally have also been greater for non-White minorities. As of 24 June 2022, for AI/AN, Black, and Hispanic/Latino groups, the rates of hospitalization were 3.0, 2.3, and 2.2 times higher than that of White Americans, respectively. The mortality rate of AI/AN, Black, and Hispanic/Latino persons was 2.1, 1.7, and 1.8 times greater than that of White Americans, respectively [2]. Consequently, though the COVID-19 pandemic has impacted the entire global population, its burden has been unequally distributed among racial and ethnic minority groups. 

New Hampshire (NH) has a small population of 1.4 million, and its residents are among the least racially and ethnically diverse in the U.S. [3]. About 87.1% of the population identifies as White alone; 1.4% Black or African American alone; 2.6% Asian alone, 4.3% Hispanic or Latino, and 4.6% other or multicultural [3]. By July 2020, NH specific data identified that Hispanic/Latinos and Black or African Americans were disproportionately more likely to get COVID-19 (1.6 and 3.9 times their population proportion, respectively), which foreshadowed what was eventually demonstrated in national data [2,4]. 

The development of safe and effective vaccines for COVID-19 offered the most important strategy to reduce the impact of the pandemic and to address this disproportionate impact. In anticipation of the availability of vaccines and that demand would greatly outstrip supply in the initial stages of vaccine rollout, the Centers for Disease Control and Prevention (CDC) required all state health departments to develop and submit a COVID-19 vaccination plan [5]. The decisions regarding allocation of these vaccines presented unprecedented challenges due to uncertainties regarding timing for vaccine approval; vaccine production, delivery and storage requirements; public lack of confidence in a novel vaccine platform; and dynamic recommendations for prioritization from various governmental and non-governmental agencies. Prioritization strategies were closely scrutinized and sometimes controversial. 

This report describes NH’s unique approach to ensuring equity in development of its vaccine allocation strategy during the time before vaccine became widely available (September 2020–April 2021). This includes the process of determining how many doses of vaccine would be distributed to which populations and when. This report presents the operational, ethical, and procedural challenges encountered, and how these were addressed. Ultimately, the goal of this report is to inform future jurisdictional pandemic response planning to protect the most vulnerable members of the population.

## 2. Methods & Approaches towards Developing the State Vaccine Allocation Strategy

### 2.1. Organization to Create the NH Vaccine Allocation Strategy 

To prepare for vaccine allocation, the NH Immunization Program formed the Vaccine Operations Section within the NH Department of Health and Human Services (DHHS) Division of Public Health Services (DPHS) Incident Management Team (IMT), which was activated in February 2020 in response to the COVID-19 pandemic. The DPHS IMT operated within the larger State of NH emergency response infrastructure, which was led by a Unified Command consisting of DHHS, the Department of Safety’s Homeland Security and Emergency Management, and the NH National Guard. The Vaccine Operations Section created the Vaccine Allocation Strategy Branch (VASB) to develop the state’s vaccine allocation strategy. There were four functional units within VASB: Ethics and Equity, Clinical Issues, Special Populations, and Data (Figure 1). In January 2021, the Ethics and Equity Unit of VASB was promoted to branch status, because it became clear that NH’s commitment to distribute and administer 10% of vaccine supply as a dedicated equity allocation was a massive undertaking. For external perspective, accountability, and independent oversight, the VASB presented preliminary decisions to the NH State Disaster Medical Advisory Committee (SDMAC). SDMAC had been established pre-pandemic to provide guidance to State Government and healthcare stakeholders during any crisis, and was comprised of State officials, legal and clinical subject matter experts, risk management professionals, community representatives, ethicists, and health leaders [6].

### 2.2. The VASB Approach to Establish an Equitable Vaccine Allocation Strategy

The VASB was charged with supporting the primary goal of the NH COVID-19 Vaccination Plan to decrease disease burden and ensure NH citizens remain healthy and free from disease in every stage of life. To achieve this, the VASB consulted multiple resources that had been developed before and during the pandemic, and ultimately declared foundational principles primarily adapted from among the ethical and procedural principles described in the National Academies of Science, Engineering, and Medicine’s (NASEM) A Framework for Equitable Allocation of Vaccine for the Novel Coronavirus [7]. These foundational principles were delineated into ethical and procedural principles and are detailed in Table 1.

The VASB then applied these principles to choose the criteria that would guide decisions regarding priority populations during each phase of vaccine allocation. Initially, the VASB adopted the allocation criteria endorsed by NASEM: risk of acquiring infection, severe morbidity and mortality, negative societal impact, and transmitting infection to others. During the review process by the Vaccine Operations Section, the SDMAC, and State Government leadership, these criteria were revised to include only the single criterion of risk of morbidity and mortality. This criterion coupled with the foundational principles functioned as the ethical framework for the VASB.

The CDC proposed that states create plans for three phases of vaccine availability (Table 2) [5]. Due to the State’s foundational criterion for reducing risk of morbidity and mortality and earlier focus on equitable allocation, VASB elected adaptations from this guidance (Figure 2).

To identify the communities at highest risk for disproportionate impact of COVID-19 because of a limited ability to mitigate, treat, and prevent transmission of a pandemic disease, NASEM recommended that states employ a social vulnerability index (SVI). VASB elected the COVID-19 Community Vulnerability Index (CCVI) [10]. The CCVI combines indicators specific to COVID-19 with the CDC SVI, which measures the expected negative impact of disasters of any type. However, the CCVI predicts only community impact and was not designed to predict individual morbidity and mortality risk. To incorporate this latter risk, VASB utilized a hybrid person- and place-based model that identified at the census tract level both (1) the geographic locations that CCVI categorized as communities at risk for negative impact; and, (2) populations that local and national data showed to be at disproportionate risk of COVID-19 infections, hospitalizations or deaths (Figure 3). This dual approach addressed both community and individual risk. 

Beginning in NH’s Phase 1b of vaccine allocation, and continuing through subsequent phases, 10% of the vaccine supply in each weekly allocation was reserved for distribution to disproportionately impacted and highly vulnerable populations. NH defined these populations as those who resided in highly vulnerable census tracts (see Figure 3), using the CCVI and US Census data for population estimates, and/or met one of the eligibility criteria listed below that elevated risk of COVID-19 infection, morbidity, and mortality. These persons were eligible to receive doses from the equity allocation, regardless of which phase they would otherwise be assigned. The equity allocation was intended to reach the most vulnerable populations in NH and to speed up their access to life-saving vaccines. This allocation ran concurrently but separately from other state vaccine allocations and utilized a different standard for eligibility. To help vaccine providers enact these recommendations, the VASB implemented “Guidelines for Equity Allocation” [11]. A person could receive a COVID-19 vaccine under the equity allocation if they met any of the following criteria: Resided in highly vulnerable census tracts (see Figure 3) using the CCVI and US Census data for population estimates; and/orMet one or more eligibility criteria regardless of residence:
○Identified as a racial and ethnic minority (all persons identified as other than White, non-Hispanic)○Were experiencing homelessness (sheltered and unsheltered)○Qualified as low income (household income at or below 185% of the poverty level)○Reported being geographically isolated or encountering physical or other barriers to travel (i.e., limited access to transportation) to points of vaccine distribution○Were homebound including:
The person’s doctor believed that their health or illness could get worse if they left the home;The person required the help of another person and/or medical equipment to leave the home, or found it difficult to leave the home and typically could not do so except for medical appointments or treatment or for short periods of time or for special non-medical events
○Reported lacking a medical home through which to verify their medical vulnerability○Experienced language/communication access barriers that prevented them from understanding vaccine registration instructions and assent during the documentation process○Reported other significant barriers that prevented them from being vaccinated through other mechanisms

Enacting these guidelines was sometimes challenging during times of extremely limited supply and high demand. Therefore, VASB provided a more detailed guide for vaccine administrators to efficiently, systematically and fairly identify those with disproportionate risk (Appendix A).

VASB also recommended to reserve a minimum of 1000 vaccine doses at all times throughout vaccine distribution, which was distinct from the 10% equity allocation. This was set aside for rapid emergency deployment to “hot spots”, defined as areas of epidemiological concern, and to remedy any errors or oversights in distribution or scheduling. This latter reason was the most common use of this vaccine reserve, which ultimately served to maintain smooth operations and ensure public trust.

### 2.3. Resources Needed for Developing an Equitable Vaccine Allocation Strategy

Given the extraordinary public health emergency presented by the COVID-19 pandemic, the rapid development of the equitable allocation strategy was an unprecedented public health accomplishment. In just one month, the VASB was established and a preliminary vaccine allocation strategy was developed. Over the next three months, the VASB navigated leadership and stakeholder approvals that led to the final public vaccination plan (Figure 2), with accompanying technical assistance documents (Appendix A). Over the 8 months that VASB met, staff dedicated hundreds of hours of labor primarily by 28 staff members from various programs (Appendix B) through 29 branch and more than 60 working group meetings. This accomplishment is even more remarkable, because, like most state health departments, public health staff expertise was often non-redundant and VASB members were required to contribute to other vital pandemic response roles as well.

### 2.4. Partner and Community Consensus on Allocation

Successful equitable vaccination is completely dependent on community engagement. NH DPHS adopted best community engagement practices recommended by NH’s COVID-19 Equity Response Team [4]. These practices included transparent, timely communication and targeted, culturally responsive messaging and outreach, and were largely dependent on strong, pre-existing community partnerships. For example, one of the most important partnerships was the COVID-19 Equity Taskforce, a public-private collaborative that was convened in March of 2020 and continues to operate in response to the community needs arising from the COVID pandemic. The taskforce provides information, establishes and strengthens partnerships, and provides a mechanism for community advocacy. By leveraging this and other partnerships and hearing community voices directly through feedback loops such as listening sessions and community meetings, VASB was able to better understand and identify community needs. These relationships additionally provided a means for community members to ensure that the state strategy addressed issues of equity through vaccine allocation. 

## 3. Results: COVID-19 Vaccine Allocation Challenges

### 3.1. Unanticipated Consequences of the Equitable Allocation Strategy

While vaccine supply was still low and demand was high, DPHS announced a priority allocation for persons disproportionately impacted by COVID-19, including racial and ethnic minorities. Some expressed opinions that this allocation was ‘not fair’ for those not in those populations. The vaccine allocation strategy was challenged through a lawsuit that named members of VASB as defendants and required members to dedicate extensive time writing affidavits, providing documentation, and testifying in court. Putting any group or person to the head of any line for critical limited health resources will be highly scrutinized and potentially contentious, which can even lead to legal challenges, funding blocks, public outcry and disabling loss of community trust.

### 3.2. Allowing for a Dynamic Allocation Strategy

Changes and delays in finalizing the equitable allocation strategy understandably can engender frustrations, among both internal and external stakeholders. One such example is that within the development process, State Government leadership necessarily reviewed and approved all aspects of the vaccine allocation strategy. This process predictably resulted in some changes, related to dynamic federal recommendations, emerging science, state resource constraints, broad stakeholder input, and varied opinions on best approach. Emerging features of the pandemic also resulted in modifications to the strategy, such as recognizing high morbidity and mortality in NH residents of long-term care facilities (including skilled nursing and assisted living facilities). As a result, VASB modified a draft version by elevating this group from Phase 1b to Phase 1a.

Based primarily on NASEM recommendations, VASB endorsed a draft allocation strategy that prioritized the population of all incarcerated and detained persons (IDP) in Phase 2. Through the review process, IDPs were not ultimately considered to be at uniform risk for morbidity and mortality and therefore not prioritized in the final strategy. Any changes in draft plans (critical for transparency) may result in the need for dedicated communication and outreach to community stakeholders.

Changes introduced during the review process sometimes resulted in additional discussions, negotiations and revisions which were resource intensive and introduced delays in strategy adoption. Early public communication highlighting the fluctuating nature of vaccine allocation strategy may have assisted in navigating stakeholder expectations and ameliorated negative reactions, especially when rationale for changes were not immediately clear.

### 3.3. Anticipating Uncertainties during Planning for Equitable Vaccine Allocation

In the early phase of the emerging COVID-19 pandemic, many uncertainties undermined confidence in the decisions regarding equitable vaccine allocation, as shown in Table 3. Strategies to mitigate this included reliance on a strong ethical framework, regular culturally relevant communications, and extensive funding and resources to address logistical challenges. During the inevitable next public health emergency, public communication strategies should include early messaging that emergency response is inherently dynamic, particularly to navigate expectations that guidance will undergo modifications and revisions. Historically underfunded and inadequate public health data systems presented an immediate challenge and significant uncertainty to equitable vaccine allocation. These systems could not readily identify populations with inequities, because data related to demographic indicators, particularly race and ethnicity, were often either unavailable or unreliable [12]. These data limitations arose from inattention to point of contact data collection (i.e., not capturing race and ethnicity appropriately at the time of service) and because some data systems were not even designed to capture these data.

### 3.4. Operational Challenges during Development of the Equitable Vaccine Allocation Strategy

In any public health emergency, it must be clear both who has the authority to make and communicate decisions. For vaccine allocation decisions in NH, the Governor’s Office and DHHS leadership held final decision-making authority, and various entities throughout the entire chain of command (including VASB) were then responsible for communicating these decisions. Open fora between stakeholders and VASB were integral to allocation planning, and many individuals and groups requested that the VASB prioritize specific persons or populations for early vaccination. These requests came through many mechanisms, such as direct communication with VASB members, non-VASB DHHS staff, external partners or State Leadership such as the Governor’s Office. The VASB met at least twice weekly to review these requests. Sometimes those answering these requests provided a preliminary decision which contradicted the final plan, which created confusion, frustration and ultimately could have undermined the public trust. Resolving these discrepancies led to the need for dynamic conflict resolution, revisions, and amendments to prioritization guidance.

Federal recommendations (e.g., from CDC) usually serve as a guide for state jurisdictions to adapt to their own context, as was the case during initial vaccine allocation where federal bodies made recommendations but states had final decision-making authority. During the COVID-19 pandemic, this was not well understood; improvd federal and state communication that each jurisdiction was empowered to decide its own priority populations and vaccine allocation strategy based on each states data and unique needs may have helped navigate expectations. For example, the ACIP recommended vaccination for essential workers within phases 2 and 3. In NH, estimates showed approximately 210,000 essential workers would be included in NH’s Phase 2—a substantial proportion of the state population. Ultimately, New Hampshire chose to prioritize vaccination of medically vulnerable and other disproportionately impacted populations instead of the broad category of all essential workers, a decision that conflicted with federal guidance [14]. Improved, pre-emptive public communication that each state jurisdiction was empowered to decide its own priority populations for vaccine allocation in accordance with emerging data may have helped to further public trust in the process.

Regarding emerging data and dynamic response, NH uniformly applied the principle that vaccine prioritization (besides the 10% equity allocation) be based exclusively on preventing morbidity and mortality and maintaining healthcare function. However, as the pandemic evolved and its broad impacts clarified, it was recognized that maintaining societal function was increasingly dependent on keeping schools open, allowing parents to remain in the workforce and continuing children’s education, as well as maintaining children’s social and emotional health. When it was publicly announced that the allocation strategy would include healthy teachers and childcare workers ahead of persons with moderate risk of morbidity and mortality, other sectors of essential workers sought inclusion.

Clinical expertise was critical to inform public health decisions during vaccine allocation strategizing. For example, most populations are diverse in terms of their medical vulnerabilities, so creating vaccination priorities among them required frequent clinical consultation. This subject matter expertise can be available through dedicated public health staffing or committed ad hoc external partnerships with clinicians. However, the pool of available ad hoc clinicians was limited as primary job responsibilities required them to provide clinical care for hospitalized patients.

Regarding the diversity among medically vulnerable individuals, one challenge encountered during decision-taking for equitable allocation were calls for prioritizing persons with intellectual disabilities. The VASB proposed and SDMAC affirmed inclusion of this population in Phase 1b. However, by the time final clearance was obtained, this approach was revised, because it became clearer that not all persons within this population had medical vulnerability. There was a process in place for healthcare providers to individually certify patients as being at significantly higher risk within Phase 1b or allow vaccine access through the equity criteria.

## 4. Discussion: Preparing for the Next Pandemic

The COVID-19 pandemic further exposed existing health disparities, often widening the gaps along the way. Jurisdictions must be better prepared to mitigate the impact of these disparities on the most vulnerable as we face the inevitable next public health crisis. The processes and outcomes in NH toward equitable vaccine allocation present an opportunity to envision a shifting paradigm for public health, focused on capitalizing on a few key lessons learned that can provide a roadmap for a more prepared and more equitable future response.

Adequate resources such as infrastructure, funding, and workforce must be available for public health institutions to equitably allocate resources in a future pandemic. Current public health infrastructure often lacks the necessary data infrastructure systems and expertise to fully engage in equity during a complex emergency response. Jurisdictions must be funded and encouraged to employ a workforce with specific skills related to health equity and then embed that expertise into response structures such as an Incident Management Team. In NH, access to an Equity Subject Matter Expert (SME) and an equity taskforce was integral to pursuing equity in all aspects of the response.

Use of vulnerability indices and access to timely, local data was critical to equitable vaccine allocation during the COVID-19 pandemic [15]. Instilling equity into both product and practice requires public health systems to think broadly about the concept of data. Public health has been traditionally reliant on data that may not produce the complete picture or may even produce a false narrative [16]. When that narrative is used to inform distribution of scarce and critical resources, inequities may further widen. In particular, jurisdictions need quality demographic data to address inequities in access to services [17]. Training for point of contact data collectors is required to increase the validity and completeness of race and ethnicity and other demographic indicator datasets. There must also be staff expertise and ability to perform disparity analyses throughout a dynamic emergency like COVID-19. In preparation for the next unique public health need, both national and localized public health data infrastructure systems will require expanded resources in the form of human capital and expertise to capture data in real time.

While the necessity of community engagement for equitable public health planning is well-known and researched, the NH experience further illustrates that the importance of community engagement, in tandem with pre-planning and transparency, cannot be overstated in emergency pandemic response [18]. In NH, we augmented national recommendations that stressed, even before the planning process began, that “effective, authentic, and meaningful engagement with community-based organizations is crucial” to not only identify priority populations but also to “build effective vaccine delivery systems that are convenient for priority populations” [13]. Broad partnerships, such as the COVID-19 Equity Taskforce, are key to addressing disparities within planning, but the middle of a crisis is not the time to begin to build new and trusting relationships. I deally these partnerships are cultivated well in advance of a future emergency. 

Education regarding the authority of federal guidance and a clearly delineated chain for external stakeholder communication before delivering approved, consistent talking points are important towards building transparency and public trust during equitable vaccine allocation. In future public health emergencies, communication should be coordinated and also delivered with messaging that strategies will change during a dynamic emergency. Communications experts can assist with describing rationales, and insertion of conditional language into prioritization announcements that may help prevent anger and disappointment when plans necessarily change. Since all groups did not have equal advocacy, the influence of advocacy could itself exacerbate inequity, and decisions should be made on the basis of medical and epidemiologic evidence as objectively as possible.

NH’s longstanding priority and processes for community engagement held vaccine allocation plans accountable to the established ethical framework, and resulted in actionable input with which to modify the allocation plan. The unexpected challenges described for reviewing requests for vaccine prioritization led us to consider how advocates and public health systems can work together to build equitable representation and ultimately access. Methods and tools that we suggest for a future response include regular listening sessions, open public fora, prepared instruments for just-in-time community assessments, small stakeholder group presentations, and defined mechanisms for the public to communicate with public health staff. In these endeavors, public health staff should utilize principles of effective risk communication, such as communicating consistently and clearly, acknowledging uncertainty, and managing expectations without dismissing concerns [7]. As a part of community engagement, NH found it beneficial to anticipate questions with job aids, tables, reminders, and “Frequently Asked Questions”. This helped NH lessen the burden of duplicated communications; enabled effective, efficient and transparent communication; and reduced risk of providing discordant information, last minute urgencies that stretched already stressed staff, and reactive plan revisions.

Equitable allocation planning requires operational understanding of implementation and distribution, as well. Shortcomings of implementation and distribution are of high consequence and may compound existing inequities. For example, nationally, eligible healthcare facilities in rural areas, counties in the top quintile for COVID-19 mortality, or counties with >42.2% non-Hispanic Black population were significantly less likely to serve as COVID-19 vaccine administration locations during early vaccine distribution [19]. As part of after action activities, each jurisdiction should incorporate disparity analysis to identify inequities in vaccine administration and distribution.

Public health jurisdictions should preemptively seek legal review of evolving allocation plans. The lawsuit, right to know and other media requests required significant time and resources from VASB subject matter experts. State health departments should be cognizant of the potential for dispute through various social, political, and legal avenues and work preemptively to predict these challenges and develop educational resources, talking points, and technical assistance guidance to mitigate disputes. Throughout the process of defending vaccine allocation plans, VASB learned of the critical importance of transparency, evidence-based decision making, and clear documentation. Having leaders with relevant expertise who are well-versed in the evidence and able to speak to allocation plan rationale was key to overcoming these challenges.

In NH, we witnessed that emergency resource allocation required continuous attention to equipoise among competing interests. Throughout the process of coordinating governmental consensus on vaccine allocation, disagreements are inevitable so negotiations are required. In these negotiations, having a foundational ethical framework to reference is key to maintain consistency and fairness.

## 5. Conclusions

The COVID-19 pandemic tested preparedness and resource allocation plans in every jurisdiction. This emergency has taught public health jurisdictions that, to control a disease that disproportionately impacts vulnerable populations, equity must be central in plans to allocate scarce resources. NH’s process of developing an equitable vaccine allocation strategy provides valuable lessons, particularly given public health resource limitations in a dynamic situation. Dedication to equity is a core function of public health and should be prioritized by every jurisdiction allocating limited resources, which requires operationalizing equity as a value, an action, and an ethical framework. Equitable allocation requires resources– both human capital and sustained funding. Well-established and trusting partnerships are key to early and focused action, which is critical to addressing disparities. Planning for equitable resource allocation for the next public health emergency should already be underway. By embracing these lessons learned, jurisdictions will be better prepared to equitably allocate life-saving resources when the next pandemic arrives.

## Figures and Tables

**Figure 1 vaccines-10-01421-f001:**
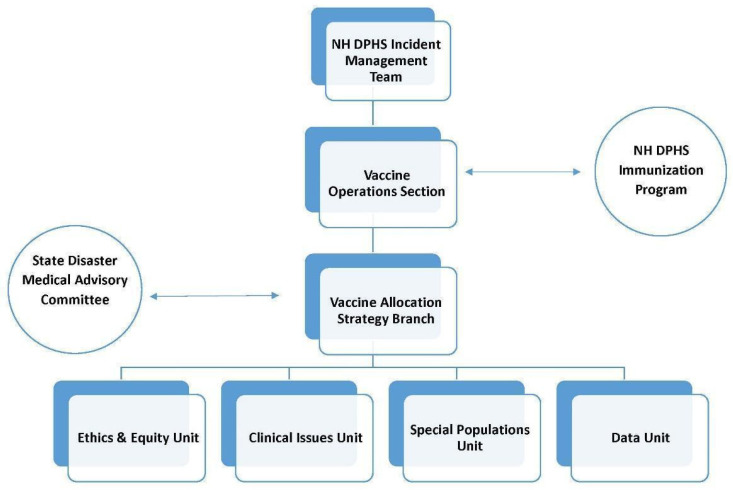
Coronavirus disease 2019 (COVID-19) Vaccine Response Organogram (September 2020). The organogram illustrates only the vaccine allocation segment relevant to the incident command structure and does not show the complete structures of the State of New Hampshire (NH) COVID-19 response, the NH Division of Public Health Services (DPHS) response, and the Vaccine Operations Section.

**Figure 2 vaccines-10-01421-f002:**
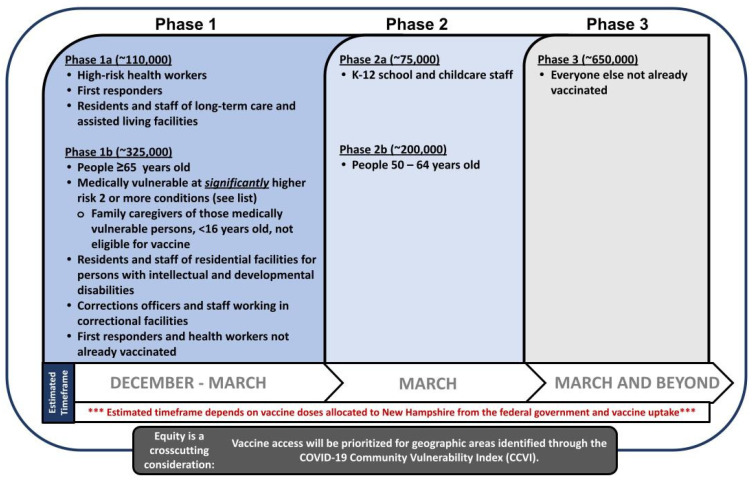
Final NH Vaccine Allocation Plan (25 March 2021) [8,9]. The plan included three phases with a crosscutting 10% Vaccine Equity Allocation beginning in Phase 1b as represented by the dark gray bar.

**Figure 3 vaccines-10-01421-f003:**
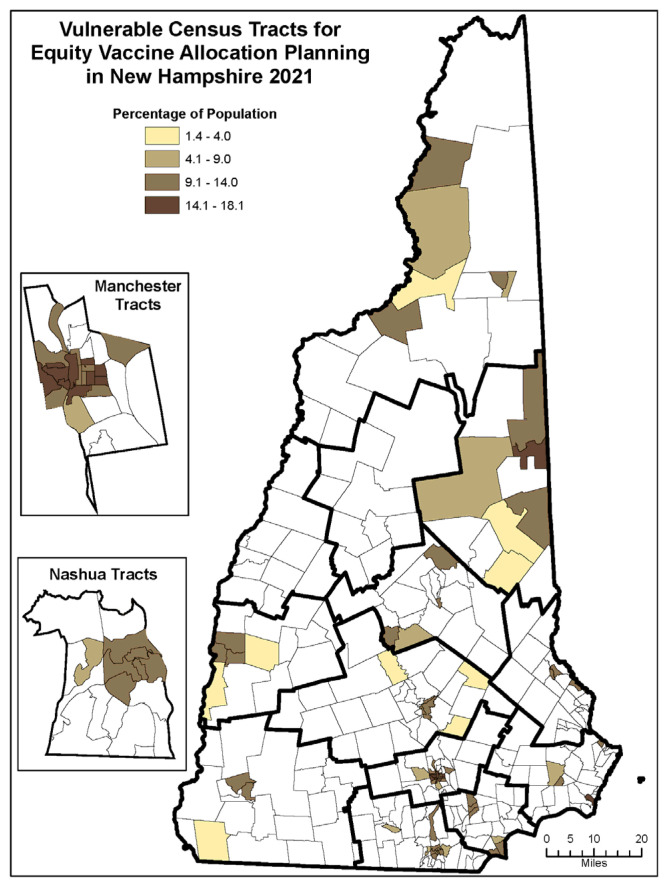
Vulnerable Census Tracts for Equity Vaccine Allocation Planning in NH. The VASB utilized the COVID-19 Community Vulnerability Index (CCVI) and Census data for population estimates to identify the communities at highest risk for disproportionate impact of COVID-19. Manchester and Nashua, the most populous cities in NH, are highlighted on the left. Only the top quartile of most vulnerable census tracts in NH are shown in the figure above.

**Table 1 vaccines-10-01421-t001:** Foundational principles of equitable allocation reprinted/adapted with permission from the National Academies of Sciences, Engineering, and Medicine (NASEM) [7]. Copyright 2020, National Academy of Sciences. The Vaccine Allocation Strategy Branch (VASB) utilized these principles for the NH COVID-19 vaccine allocation framework.

Foundational Principles	Definition
Ethical Principles:	
Maximum benefit	Encompasses the obligation to protect and promote the public’s health and its socioeconomic well-being in the short and long term
Equal concern	Requires that every person be considered and treated as having equal dignity, worth, and value
Mitigation of health inequities	Includes the obligation to explicitly address the higher burden of COVID-19 experienced by the populations affected most heavily, given their exposure and compounding health inequities.
Procedural Principles:	
Fairness	Requires engagement with the public, particularly those most affected by the pandemic, and impartial decision making about and evenhanded application of allocation criteria and priority categories.
Transparency	Includes the obligation to communicate with the public openly, clearly, accurately, and straightforwardly about the allocation framework as it is being developed, deployed, and modified.
Evidence-based	Expresses the requirement to base the allocation framework, including its goal, criteria, and phases, on the best available and constantly updated scientific information and data.

**Table 2 vaccines-10-01421-t002:** Centers for Disease Control and Prevention (CDC) recommended three-phase allocation strategy, reflecting supply and demand of COVID-19 vaccine [5].

Phase	Predicted Supply & Demand	Target/Focus
1	Severely limited doses available	Those at highest risk of morbidity and mortality and frontline healthcare workers
2	Large number of doses available, supply likely to meet demand	Additional high-risk populations not vaccinated during phase one
3	Likely sufficient supply, slowing demand	Equitable vaccination access

**Table 3 vaccines-10-01421-t003:** Factors Affecting Equitable Vaccine Allocation for NH. This table, adapted from NASEM for the NH context, shows challenges associated with early pandemic COVID-19 vaccine allocation, and how these factors affected equitable vaccine allocation [13].

Features of Early Pandemic Vaccine Allocation	Implications for Equity
Uncertain number and timing of available vaccine doses	● The inability to commit to vaccine allocation in advance likely disproportionately negatively impacted persons who qualified for equity doses, because they often require more time to plan and access vaccine than those who do not (e.g., finding childcare, arranging transportation, planning day off work) ● Messaging that described tentative vaccination availability could undermine already fragile trust in government public health systems among populations requiring equitable distribution, particularly among historically marginalized populations
Uncertain vaccine efficacy, especially for vulnerable populations	● Community partners sometimes expressed preference for the more operationally feasible but less efficacious single-dose Johnson and Johnson / Janssen vaccine, which may have resulted in disproportionate allocation for transient and high-need populations (e.g., migrant, incarcerated, homeless and homebound) ● Differences in feasibility/efficacy required careful balance between public demand and public benefit, including the decision to not direct only one vaccine product based on logistical considerations (e.g., challenges with reaching transient populations for second doses)
Vaccine safety, overall and in different populations	● Communicating incomplete vaccine safety data may have increased vaccine hesitancy, which was initially higher among populations requiring equitable distribution, particularly among historically marginalized populations ● Development of clear messaging regarding safety was delayed for certain populations, particularly those requiring non-English and plain language messaging
Extreme cold chain requirements for the mRNA vaccines	● Some groups requested specific vaccine based on ease of handling and storage capacities. Sites serving lower resource communities were less likely to have the logistical capability to handle all storage and handling requirements, resulting in less vaccine access to the highest need populations ● Unanticipated resources were required to maintain cold chain, including appropriate refrigerators/freezers and dedicated mobile providers
Social, economic, and legal contexts	● Considerable resources were required to balance and communicate differential risk that resulted in the prioritizing of some populations over others ● Equitable allocation practices should incorporate a public input process, including a mechanism for requests for prioritization, and transparency of final decisions and the decision-making process

## Data Availability

No new data were created or analyzed in this study. Data sharing is not applicable to this article.

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
