# Peer review of "Ensuring Equitable COVID-19 Vaccine Allocation in New Hampshire: The First Eight Months toward a New Era"

_vaccines, 2022, doi:10.3390/vaccines10091421_

Round 1
Reviewer 1 Report
The present study "Ensuring Equitable COVID-19 Vaccine Allocation in New 3 Hampshire: The First Eight Months toward a New Era" is to serve as an example and model for other cities and communities how to allocate vaccines in case of pandemic. The lessons learned and described here will facilitate more nimble 25 and equitable jurisdictional responses in future public health emergencies.
The paper is well structured, discussed and documented with all the data on the manuscript or on the supplementary material.
One of the weaknesses that I can identify more references to support the paper.
Reviewer 2 Report
Authors did excellent job to ensure equity in development of the New Hampshire vaccine allocation strategy prior to the availability of vaccine (September 2020 – April 2021). Their approach includes number of dose, distribution and categorizing with exposure to virus and deciding when to administer the vaccine dose. they eloquently reported the limitations and challenges encountered and explained the multilayered strategies to overcome present and future challenges. This report will possibly help to articulate and plan for possible future pandemic response.
I have some minor comments:
Authors may elaborate this statement further.
"In July 2020, Hispanic/Latinos and Black or African Americans appeared disproportionately likely to get COVID-19 (1.6 and 3.9 times their population proportion, respectively) (Governor’s COVID-19 Equity Response Team, 2020)." Line 51-54
Please add reference or elaborate this statement “ Public health has been traditionally reliant on data that may not produce the complete picture or may even produce a fully false narrative”. Line 379
Reviewer 3 Report
Please correct all references according to journal stile.
References should be described as follows:
Journal Articles:
1. Author 1; Author 2. Title of the article. Abbreviated Journal Name Year; Volume: page range.
Moreover, you are required to include in-text references in the main body of your work in square brackets [1].
